# REVEAL: Multimodal Vision–Language Alignment of Retinal Morphometry and Clinical Risks for Incident AD and Dementia Prediction

**Seowung Leem**[1]  ID                                    LEEM.S@UFL.EDU
**Lin Gu**[2] ID                                    RIN.TANI.E8@TOHOKU.AC.JP
**Chenyu You**[3,4] ID                                CHENYU.YOU@STONYBROOK.EDU
**Kuang Gong**[1] ID                                    KGONG@BME.UFL.EDU
**Ruogu Fang**[*1] ID                                Ruogu.Fang@BME.UFL.EDU

[1] *J. Crayton Pruitt Family Department of Biomedical Engineering, University of Florida, United States*

[2] *Research Institute of Electrical Communication, Tohoku University, Japan*

[3] *Department of Applied Mathematics & Statistics, Stony Brook University, United States*

[4] *Department of Computer Science, Stony Brook University, United States*

**Editors:** Accepted for publication a MIDL 2026

## Abstract

The retina provides a unique, noninvasive window into Alzheimer's disease and dementia, capturing early structural changes through morphometric features, while systemic and lifestyle risk factors reflect well-established contributors to AD and dementia susceptibility long before clinical symptom onset. However, current retinal analysis frameworks typically model imaging and risk factors separately, preventing them from capturing the joint multimodal patterns that are critical for early risk prediction. Moreover, existing methods rarely incorporate mechanisms to organize or align patients with similar retinal and clinical characteristics, limiting their ability to learn coherent cross-modal associations. To address these limitations, we introduce REVEAL (**RE**tinal-risk **V**ision-language **E**arly **A**lzheimer's **L**earning) that aligns color fundus photographs with individualized disease-specific risk profiles for incident AD and dementia prediction on average 8 years before diagnosis (range: 1–11 years). Because real-world risk factors are structured questionnaire data, we first translate them into clinically interpretable narratives compatible with pretrained vision-language models (VLMs). We further propose a group-aware contrastive learning (GACL) strategy that clusters patients with similar retinal morphometry and risk factors as positive pairs, strengthening multimodal alignment. This unified representation-learning framework substantially outperforms state-of-the-art retinal imaging models paired with clinical text encoders, as well as general VLMs, demonstrating the value of jointly modeling retinal biomarkers and clinical risk factors. By providing a generalizable, noninvasive approach for early AD and dementia risk stratification, REVEAL has the potential to enable earlier interventions and improve preventive care at the population level.

**Keywords:** Retinal morphometry, risk factors, Alzheimer's disease and related dementia, Vision-language alignment, Contrastive learning

---

* Corresponding Author

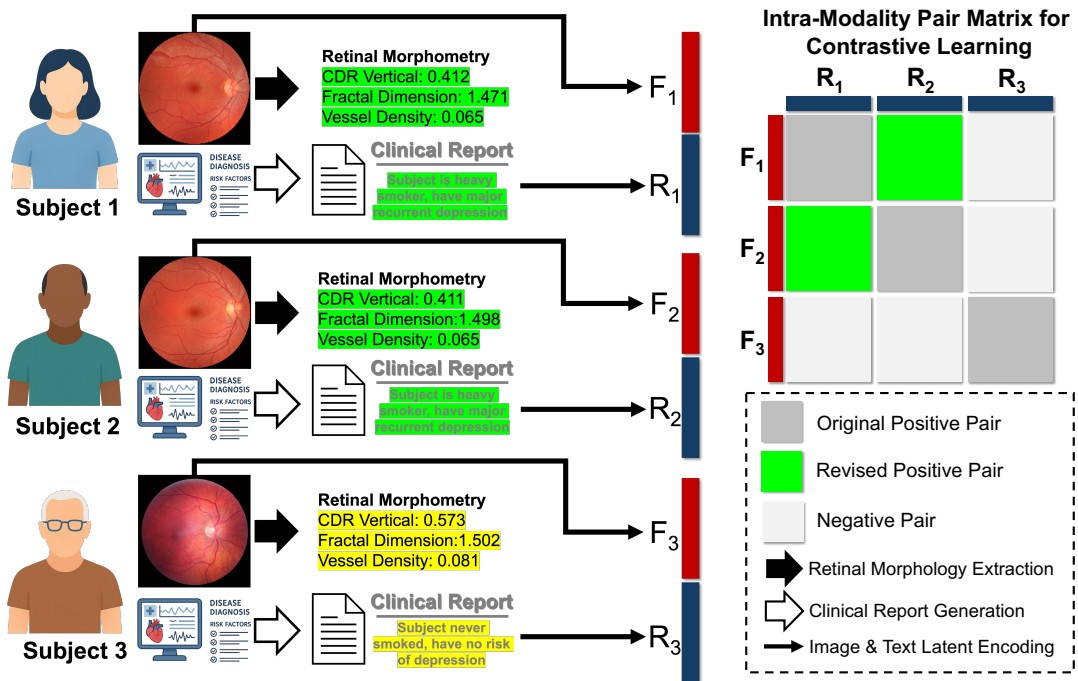

Figure 1: Schematic of clinical scenario and proposed method.

## 1. Introduction

Alzheimer's disease and dementia are progressive neurodegenerative diseases that manifest years before clinical symptom onset. Early identification of individuals at risk is critical for timely intervention and prevention. The retina offers a unique, noninvasive window into AD and dementia. Retinal morphometric features, referring to a set of quantitative measurements characterizing the size, shape, and structure of retinal components, have been shown to reflect early neurodegenerative changes and amyloid-$\beta$ or tau deposition in the brain (Cheung et al., 2021; Koronyo et al., 2017; Ravichandran et al., 2025; Byun et al., 2021; Snyder et al., 2016). Parallel to retinal alterations, AD and dementia risk is strongly influenced by systemic and lifestyle factors (Leshner et al., 2017; Sprecher et al., 2017; Xiong et al., 2023; Hayden et al., 2024; Huszár et al., 2024; Livingston et al., 2024). While retinal morphometry captures early neurodegenerative signatures, risk factors provide complementary information on modifiable factors that contribute to disease susceptibility. This convergence suggests that jointly modeling retinal biomarkers and systemic risk factors could improve early AD and dementia prediction beyond what either modality can achieve alone.

Despite this potential, current approaches typically analyze retinal images and risk factors separately, limiting their ability to capture the complex multimodal relationships underlying preclinical AD and dementia. Conventional contrastive learning frameworks often fail to align patients who share both retinal and systemic risk characteristics, leading to overlooked clinical commonalities (Figure 1). Moreover, structured risk-factor data from questionnaires cannot be directly incorporated into standard vision-language models (VLMs), which are pretrained on natural language, creating a modality gap.

To address these challenges, we introduce **REVEAL** (**RE**tinal-risk **V**ision-language **E**arly **A**lzheimer's **L**earning), a novel VLM-based framework that integrates retinal morphometric features with individualized disease-specific risk profiles. Structured risk factors are first transformed into clinically meaningful narratives using large language models, enabling seamless multimodal representation learning. We further propose a **group-aware contrastive learning strategy** that leverages intra-modality similarity to identify clinically aligned individuals, capturing shared pathophysiological patterns across subjects. This approach allows REVEAL to learn unified representations that more accurately reflect the interplay between retinal biomarkers and systemic risk factors, offering improved early AD and dementia risk stratification. Our work has the following contributions:

- We introduce **REVEAL**, the first framework to jointly model fundus images and individualized AD and dementia risk factors by translating structured questionnaires into clinically meaningful narratives compatible with pretrained VLMs.

- We propose a **group-aware contrastive learning strategy** that identifies subjects sharing similar retinal morphometry and risk profiles, enabling coherent and clinically aligned multimodal representation learning.

- REVEAL achieves state-of-the-art performance in predicting incident AD and dementia on average 8 years before clinical onset (AD: mean = 8.68 years, range = 2.38–11.58 years; dementia: mean = 8.49 years, range = 1.50–11.58 years) over retinal-only, clinical-text, and general VLM baselines, providing a generalizable, noninvasive approach for population-level early AD and dementia risk stratification.

## 2. Method

### 2.1. Overview of REVEAL Framework

The REVEAL framework was designed to operate in two stages. First, it aligned fundus images with individualized AD and dementia risk factors using a CLIP-style contrastive learning approach with our novel image-text pairing strategy. This enabled the model to learn multimodal relationships between colored fundus photography (CFP) and biological, phenotypic, and clinical markers of preclinical AD and dementia. Second, the learned joint representations were utilized in a downstream classifier to predict incident, preclinical AD and dementia (see Section 3.1 for details).

### 2.2. Constructing Clinical Report and Group-Aware Labels for Contrastive Learning

2.2.1. Synthetic Clinical Report Generation

Direct application of CLIP was not feasible because the risk factors are represented as structured, tabular variables rather than natural-language descriptions. However, alignment of fundus images with risk factors required a shared representation space that VLM can operate on. To bridge the modality gap between structured risk-factor variables and the natural-language input required by VLMs, we synthesized standardized clinical-style narratives from tabular health data (Figure 2). This transformation enabled the VLM to interpret the tabular risk factors in a linguistically contextualized form and facilitates multimodal alignment between fundus images and clinical attributes relevant to AD and

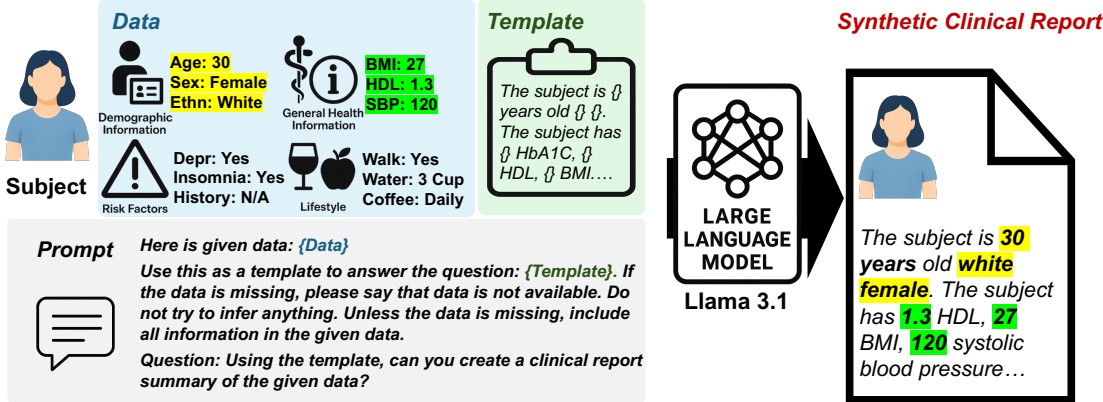

Figure 2: Schematic overview of how a synthetic clinical report is generated.

dementia. Using the LLaMA-3.1 API as the text generation engine (Grattafiori et al., 2024), we converted each participant's risk factor profile into a synthetic clinical report. For each subject, the LLM was provided with (1) a template prompt, (2) the subject's structured risk factor values, and (3) explicit instructions for generating a concise medical summary. The template was adapted from the "Patient Information" section of the CARE clinical case report guideline (Gagnier et al., 2013), ensuring that the synthesized narratives follow established clinical documentation conventions. The input prompt was designed to map the tabular information 1:1 into a template to prevent potential variability (Appendix A). This process produced consistent, clinically meaningful text representations that enable seamless integration of structured health information into our multimodal predictive framework.

### 2.2.2. Group-aware Contrastive Learning Strategy

Conventional CLIP-style frameworks often fail in the medical domain (Radford et al., 2021). Prior studies showed that naive CLIP approaches struggle to capture the complex semantic relationships between images and disease-level information, highlighting the need for domain-specific strategies (Wang et al., 2022; Eslami et al., 2023). In our context, individuals sharing both retinal and systemic AD and dementia risk characteristics must be grouped during training, since conventional contrastive learning only treats image-text pairs from the same subject as positive matches. To mitigate these gaps and enable the model to capture shared pathophysiological patterns across different modalities, we designed a group-aware contrastive learning (GACL). To introduce explicit clinical grounding, our GACL leverages morphometric features extracted directly from CFP, rather than solely on latent representations from image encoders. This addressed limitations from prior works that attempted to improve the shortcomings of conventional CLIP by introducing the image-level or latent-level similarity (Du et al., 2024; Wu et al., 2024), which lacked explicit clinical grounding to find phenomenologically similar individuals with clinical relevance. By this design, the REVEAL learns a contrastive objective that encourages the model to associate the patterns of retinal signals and risk profiles that are linked to those patterns. Thus, REVEAL learns a latent disease-risk manifold, not object-level semantics. The GACL was inspired by (Bulat et al., 2024).

As shown in Figure 3, $\mathbf{F} \in \mathbb{R}^{N \times K}$ and $\mathbf{T} \in \mathbb{R}^{N \times D}$ denote the z-normalized morphometric feature matrix and l2-normalized embeddings clinical report matrix for all $N$ samples in

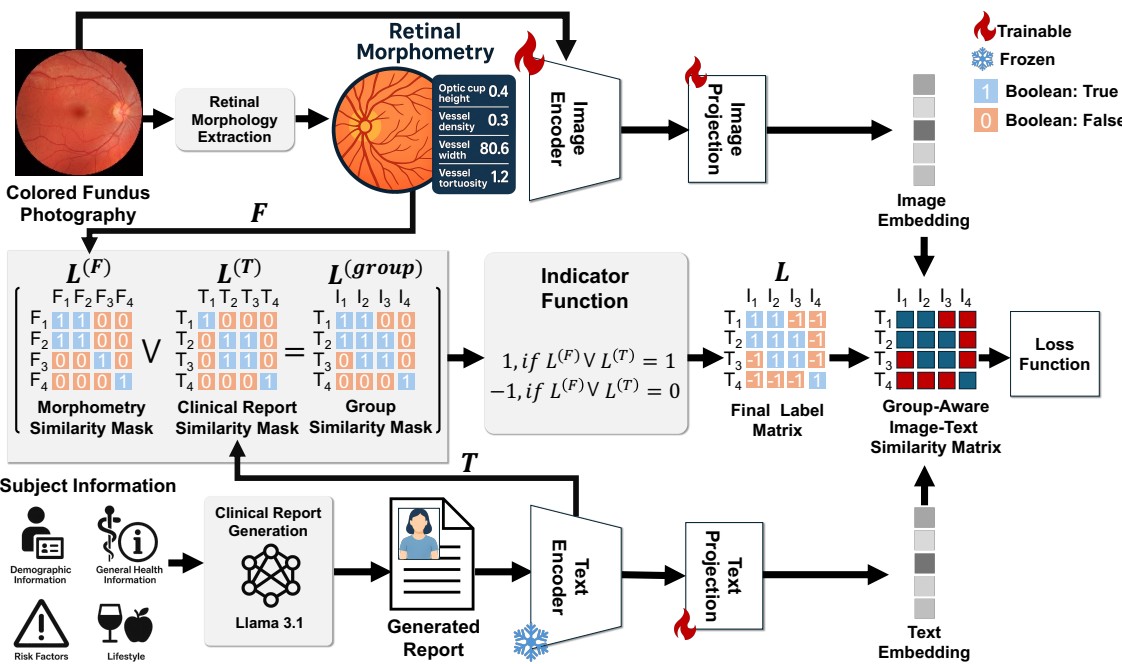

Figure 3: Schematic overview of how GACL is performed.

a training batch, respectively. Here, $K$ represents the number of morphometric features and $D$ denotes the dimensionality of the text embedding space. To quantify the pairwise relationship within each modality, we computed the intra-morphometry similarity matrices $\mathbf{S^{(F)}} \in \mathbb{R}^{N \times N}$ for the fundus image and intra-clinical report similarity matrix $\mathbf{S^{(T)}} \in \mathbb{R}^{N \times N}$ for text.

$$\mathbf{S^{(F)}} = \mathbf{F} \cdot \mathbf{F}^{\top}, \quad \text{and} \quad \mathbf{S^{(T)}} = \mathbf{T} \cdot \mathbf{T}^{\top}. \tag{1}$$

Each entry in $\mathbf{S^{(F)}}$ characterizes how similar the retinal morphometric profiles of two subjects are, with a larger value indicating closer structural resemblance. Likewise, $\mathbf{S^{(T)}}$ captures the semantic similarity between the clinical report embeddings, demonstrating the degree to which two subjects share encoded risk factor profiles. To identify subjects with similar characteristics, we thresholded both similarity matrices using modality-specific thresholds $\boldsymbol{\tau_F}$ and $\boldsymbol{\tau_T}$, yielding binary similarity masks $\mathbf{L^{(F)}}$ and $\mathbf{L^{(T)}}$. In each mask, a value of **1 (Boolean True)** indicates a similar sample pair, while **0 (Boolean False)** indicates a dissimilar pair.

$$\mathbf{L^{(F)}} = \begin{cases} 1(True), & \text{if } \mathbf{S^{(F)}} > \boldsymbol{\tau_F} \\ 0(False), & \text{otherwise} \end{cases} \quad \text{and} \quad \mathbf{L^{(T)}} = \begin{cases} 1(True), & \text{if } \mathbf{S^{(T)}} > \boldsymbol{\tau_T} \\ 0(False), & \text{otherwise} \end{cases} \tag{2}$$

To integrate information across modalities, we obtained a group similarity mask $\mathbf{L^{(group)}}$ by applying a logical OR operation between two modality-specific masks. Finally, the group similarity mask was mapped by an indicator function, resulting in a contrastive learning-compatible final label matrix $\mathbf{L}$, where entries of 1 were preserved, and 0s were converted

to -1. This formulation preserved similarity relationships across modalities, ensuring that image-text alignment benefits from both structural consistency (from morphometry) and semantic consistency (from clinical reports). By reinforcing agreement between intra-modal similarity, image-text pairings were improved to maximize the learning efficiency between retinal morphometric features and risk factors.

$$\mathbf{L}^{(\text{group})} = \begin{cases} 1, & \text{if } \mathbf{L}^{(\mathbf{F})} \vee \mathbf{L}^{(\mathbf{T})} = 1, \\ 0, & \text{otherwise} \end{cases} \quad \text{and} \quad \mathbf{L} = \begin{cases} 1, & \text{if } \mathbf{L}^{(\text{group})} = 1, \\ -1, & \text{otherwise} \end{cases} \tag{3}$$

### 2.3. Image-Text Alignment Learning with REVEAL

#### 2.3.1. REVEAL Architecture

The REVEAL framework was built on a standard contrastive vision-language learning setup to capture joint patterns between fundus images and AD and dementia risk factors. As shown in Figure 3, we used RETFound (Zhou et al., 2023) as the image encoder and GatorTron (Yang et al., 2022) as the text encoder, adding only lightweight projection layers to align their feature dimensions. During each forward pass, a raw fundus image and its synthesized clinical report were encoded and projected into a shared latent space. Retinal morphometrics and clinical narratives were further integrated into the GACL procedure to construct a label matrix. Finally, a group-aware image-text similarity matrix was computed using image embedding, text embedding, and a label matrix. The trainable REVEAL components were denoted as "flame" in Figure 3. This design enabled REVEAL to leverage both retinal imaging priors from foundation models and semantic priors from clinically trained language models.

#### 2.3.2. Contrastive learning

With GACL, the conventional contrastive objective was no longer applicable because it accommodated only a single positive pair per sample. Therefore, we adopted the loss from the prior work (Bulat et al., 2024) to support multiple clinically aligned pairs.

$$\mathcal{L} = -\frac{1}{N_{\text{img}} N_{\text{txt}}} \sum_{i=1}^{N_{\text{img}}} \sum_{j=1}^{N_{\text{txt}}} \log \left( \frac{1}{1 + \exp\left(l_{ij}\left(-s_{ij}/\tau + \beta\right)\right)} \right), \tag{4}$$

$N_{\text{img}}$ and $N_{\text{txt}}$ denote the number of images and texts in a training batch. The label term $l_{ij} \in \{+1, -1\}$ is the $(i, j)$-th entry of the final label matrix $L$, with $l_{ij} = 1$ indicating a similar (positive) image–text pair and $l_{ij} = -1$ indicating a dissimilar (negative) pair. The similarity value $s_{ij}$ is computed as the cosine similarity between the corresponding $i$-th image and $j$-th text embeddings obtained from the REVEAL framework. The temperature parameter is fixed at $\tau = 0.07$. The bias term $\beta$ is introduced to stabilize early training by reducing the initial loss, which is otherwise dominated by the large number of negative pairs. Including $\beta$, all hyperparameters (learning rate, eps, weight decay) and similarity thresholds ($\boldsymbol{\tau_F}$ and $\boldsymbol{\tau_T}$) were chosen using an Optuna, hyperparameter optimization framework (Akiba et al., 2019), which identifies the optimal configuration within user-defined search ranges (details in Appendix B).

## 2.4. Study Population and Data Preprocessing

### 2.4.1. SUBJECT SELECTION

Color fundus photographs (CFPs) and AD and dementia-related risk factors were obtained from the UK Biobank (Sudlow et al., 2015). A total of 39,242 participants with high-quality CFPs were included and allocated into training (n=30,462), validation (n=3,384), and test (n=5,396) sets (Table 1, preprocessing details in Section 2.4.2). These splits each served a distinct role within the REVEAL framework. The training and validation sets were used solely in Stage 1 for representation alignment, with the validation set guiding hyperparameter tuning and similarity-threshold selection, while the test set was reserved for Stage 2 AD and dementia prediction.

All participants who later developed incident AD or dementia were assigned to the test set, and only participants free of both prevalent and incident disease were included in the training and validation sets. Incident diagnoses were identified using UK Biobank dementia fields (42018, 42020, 42022, 42024). Among individuals with high-quality CFPs, 86 developed incident AD (mean time to diagnosis: 8.68 years; range: 2.38–11.58) and 93 developed dementia of any subtype (mean: 8.49 years; range: 1.50–11.58).

To form the final evaluation cohort, control subjects without incident AD and dementia were sampled from the test pool to achieve an approximate 12% disease prevalence, consistent with estimates for adults aged ≥65 years (Xiaopeng et al., 2025), while maintaining age and gender matched distributions (AD controls = 1,077; dementia controls = 1,139). From this cohort, 931 subjects (862 controls, 69 AD) for AD prediction and 985 subjects (911 controls, 74 dementia) for dementia prediction were used to train SVM models with 5-fold cross-validation. The remaining subjects, 232 (215 controls, 17 AD) for AD prediction and 247 (228 controls, 19 dementia) for dementia prediction, were held out as an independent test set. Cohort characteristics for downstream prediction tasks are provided in Tables 6 and 7 (Appendix C), and the distribution of onset years for AD and dementia are shown in Figure 4 (Appendix D)

### 2.4.2. RISK FACTOR COMPILATION AND RETINAL IMAGE PROCESSING

A comprehensive set of demographic, behavioral, cognitive, and lifestyle variables was compiled as risk factors based on established epidemiological evidence (Leshner et al., 2017; Sprecher et al., 2017; Xiong et al., 2023; Hayden et al., 2024; Huszár et al., 2024; Livingston et al., 2024). The full list of these risk factors are provided in Appendix E. For the CFPs, image preprocessing and retinal morphometric feature extraction were carried out using the AutoMorph fundus morphology quantification pipeline (Zhou et al., 2022). A total of 136,994 CFPs were available from the initial UK Biobank assessment visit. AutoMorph

Table 1: Demographic characteristics of the UK Biobank participants across the training, validation, and test cohorts.

|  | Train (n=30,462) | Validation (n=3,384) | Test (n=5,396) |
| --- | --- | --- | --- |
| Gender: (male %) | 45.10 | 45.41 | 45.10 |
| Age: mean (s.d) | 55.53 (8.24) | 55.78 (8.12) | 55.52 (8.17) |
| Ethnicity: (British %) | 84.08 | 83.51 | 88.51 |

first applied a convolutional neural network–based quality-control module that classified images as low, moderate, or good quality. Following automated quality filtering and subsequent manual review, 66,251 high-quality images from 39,242 participants were retained for analysis. From these curated images, AutoMorph produced a structured set of retinal morphometric features (K=17; full list provided in Appendix F). These structural features have been shown in prior research to exhibit measurable differences in both preclinical and clinical stages of AD and dementia (Frost et al., 2013; Sharafi et al., 2019; Valenti, 2011; Ong et al., 2014; Armstrong et al., 2021). To maintain consistent anatomical orientation across eyes, all right-eye images were horizontally flipped before feature extraction.

## 3. Experiments

### 3.1. Downstream Tasks

#### 3.1.1. Incident AD and incident dementia prediction

We evaluated REVEAL on two prediction tasks: incident AD and incident dementia. For both tasks, we trained a multimodal SVM with an RBF kernel to perform binary classification, distinguishing individuals who later developed AD and dementia (normal at initial baseline visit and diagnosis reported after 1-11 years after baseline) from those who remained cognitively normal. The SVM produced probabilistic outputs, providing likelihood estimates for being AD/dementia-positive versus control. Each subject was represented by a concatenated multimodal feature vector composed of L2-normalized CFP image embeddings and text embeddings extracted from the REVEAL encoders. Class-weighted training was used to mitigate the imbalance between incident cases and controls. SVM hyperparameters ($C$ and $\gamma$) were tuned using 5-fold cross-validation, and the best-performing model was subsequently evaluated on the independent hold-out test set. All reported results correspond to this final evaluation.

#### 3.1.2. Comparison models

To evaluate REVEAL, we compared its performance with several strong fundus-based foundation models: RETFound (CFP) (Zhou et al., 2023), RET-CLIP (Du et al., 2024), and KeepFIT-CFP (Wu et al., 2024), as well as medical multimodal vision-language models trained on multiple medical imaging types, including PMC-CLIP (Lin et al., 2023) and BiomedCLIP (Zhang et al., 2025). Because RETFound was an image encoder-only model, we paired it with GatorTron (Yang et al., 2022) to enable both image and text representation. In the analysis, embeddings from two models were simply concatenated. In addition to these baselines, we trained a tabular SVM using clinical variables and CFP-derived morphometric features, applying most-frequent imputation for categorical variables and median imputation for continuous variables. Specifically, we tested tabular risk factors and morphometric features and risk factors with CFP latents to evaluate whether the improvement stems from the semantic richness of the LLM narrative or simply the power of the image foundation model. All models followed the same training and testing protocol as the multimodal SVM. Each experiment was repeated 10 times with different random seeds, and we report the average performance across runs. We used Welch's t-test and Hedge's g to evaluate the statistical difference between REVEAL and comparison methods.

#### 3.1.3. Threshold Evaluation of REVEAL Framework

In REVEAL, thresholds $\tau_F$, and $\tau_T$ from GACL determine which image-text pairs should be grouped to share information, to learn shared representations among phenomenologically

similar samples. Thresholds that are too low introduce noise by aligning dissimilar pairs, whereas thresholds that are too high restrict the model's ability to capture meaningful cross-modal relationships. To assess their influence on predictive performance, we trained the model using varying threshold configurations. In each experiment, one threshold was fixed at the optimal value determined during optimization, while the other was varied systematically. Threshold candidates were chosen from the quartiles of the morphometric and text similarity distributions in the development set.

### 3.1.4. EVALUATING CLINICALLY GROUNDED SIMILARITY IN GACL

As previously noted in Section 2.2.2, prior works have attempted to remedy the shortcomings of conventional CLIP by incorporating image-level or latent-level similarity. To evaluate the contribution of clinically grounded similarity in GACL, we compared downstream prediction performance under two configurations: (1) GACL using morphometric features as the source of image-image similarity, and (2) GACL using similarity computed directly from the image embeddings produced by the image encoder. This comparison allowed us to isolate the benefit of explicit clinical grounding for identifying phenotypically similar subjects and enhancing downstream AD and dementia prediction.

### 3.1.5. EVALUATING THE EFFECT OF DIFFERENT LOGICAL OPERATORS IN GACL

In preclinical disease settings, phenotypic similarity across different modalities can emerge asynchronously. For instance, individuals may share clinical risk factors indicative of elevated neurodegenerative disease risk, while corresponding retinal signatures may not yet be present. To account for this asynchrony, GACL adopted a logical OR operator when defining group-level similarity. To validate this design choice, we conducted a comparative analysis using the logical AND operator. Specifically, we replaced the OR operator in Equation 3 with an AND operator while keeping all other parameters fixed.

### 3.2. Result

Table 2: Performance of the incident Alzheimer's Disease prediction task. The average of 10 random seeds is presented as mean±std. The best results for each modality are in bold text. See Table 8 for statistics and effect size.

|  | AUROC | Balanced Accuracy | F1-Score | MCC |
|---|---|---|---|---|
| Baseline SVM | 0.593±0.068 | 0.574±0.083 | 0.140±0.089 | 0.076±0.099 |
| KeepFIT-CFP | 0.503±0.061 | 0.519±0.041 | 0.117±0.038 | 0.018±0.045 |
| BiomedCLIP | 0.525±0.066 | 0.522±0.052 | 0.121±0.055 | 0.023±0.057 |
| RETCLIP | 0.558±0.076 | 0.527±0.042 | 0.106±0.069 | 0.028±0.051 |
| PMC-CLIP | 0.471±0.052 | 0.484±0.020 | 0.076±0.024 | -0.022±0.024 |
| RETFound+GatorTron | 0.655±0.060 | 0.573±0.057 | 0.174±0.098 | 0.108±0.095 |
| Ours (no GACL) | 0.654±0.097 | 0.602±0.078 | 0.205±0.101 | 0.144±0.111 |
| Ours (with GACL) | **0.658±0.095** | **0.610±0.083** | **0.208±0.105** | **0.147±0.117** |

### 3.2.1. GROUP-AWARE CONTRASTIVE LEARNING IMPROVES THE INCIDENT AD AND DEMENTIA PREDICTION

In the incident AD prediction task (Table 2), REVEAL achieved the best performance across nearly all evaluation metrics, including AUROC, balanced accuracy, F1-Score, and

Matthew's Correlation Coefficient (MCC). Notably, the multimodal SVM trained on RE-VEAL embeddings substantially outperformed a baseline SVM trained directly on tabular risk factors and raw retinal morphometric features, demonstrating that vision-language embeddings effectively transform raw modalities into enriched representations. Incorporating GACL further improved performance by aligning patients with similar retinal morphometry and risk profiles, enhancing overall predictive power. In the broader incident-dementia prediction task (Table 3), the SVM using REVEAL embeddings again outperformed baseline SVMs and other vision-language models. These results indicate that group-aware alignment strengthens multimodal representation learning, in both AD and dementia cases, demonstrating that retinal structural features closely correspond to disease-specific biomarkers. Statistical analysis of AD and dementia (Tables 8 and 9 in Appendix G) shows that these improvements are highly significant and associated with large effect sizes when compared to conventional CLIP-based models and SVM baselines. While comparisons with RET-Found+GatorTron do not always reach conventional statistical significance, these tests are based on 10 independent runs and are therefore underpowered to detect small-to-moderate effects. Importantly, GACL consistently improves predictive performance with non-negligible effect sizes, indicating meaningful practical gains rather than equivalence. The consistent improvements introduced by GACL highlight its effectiveness in enhancing representation learning for long-term neurodegenerative disease risk prediction. Importantly, all CFPs in embedding learning were collected from cognitively normal participants at baseline, emphasizing that REVEAL, combined with a multimodal SVM, can identify preclinical AD and dementia risk by leveraging the complementary information between retinal morphometry and systemic risk factors.

We further conducted an ablation study to examine the contribution of individual components in our model for incident AD and dementia prediction (Table 10 in Appendix H). Specifically, we evaluated the model using image embeddings alone (Image-only), image embeddings combined with raw tabular risk factors (Image+Table), and text embeddings alone (Text-only). Across both prediction tasks, Text-only representations consistently outperformed both Image-only and Image+Table variants, suggesting that clinical narratives capture substantially richer signals relevant to neurodegenerative disease risk. Notably, the joint Image-Text representation (REVEAL) achieved the best overall performance across all evaluation metrics, indicating that the enriched image representations provide complementary information beyond text alone. In contrast, the Image+Table configurations underperformed the Text-only model, despite incorporating structured clinical variables. This finding highlights the advantage of replacing raw tabular features with clinical narratives, underscoring the benefit of higher-level semantic abstractions over simple concatenation between different model features.

### 3.2.2. Impact of Thresholds on REVEAL Performance

The relative percentage differences in downstream prediction performance between the model trained with the optimal threshold and those trained under varying $\tau_F$ and $\tau_T$ in REVEAL are shown in Figure 5 of Appendix I. Compared to the performance metric from the REVEAL with the optimal threshold (gray horizontal line, where values below 0 indicate worse performance and values above 0 indicate improvement), other models trained with different image or text thresholds did not yield better performance in most cases for both AD and dementia. For AD, using the highest $\tau_F$ produced the best accuracy, F1-score,

Table 3: Performance of the incident dementia prediction task. The average of 10 random seeds is presented as mean ± standard deviation. The best results for each modality are in bold text. See Table 9 for statistics and effect size.

|  | AUROC | Balanced Accuracy | F1-Score | MCC |
|---|---|---|---|---|
| Baseline SVM | 0.571±0.092 | 0.572±0.041 | 0.151±0.042 | 0.075±0.041 |
| KeepFIT-CFP | 0.487±0.038 | 0.505±0.041 | 0.110±0.032 | 0.005±0.040 |
| BiomedCLIP | 0.487±0.043 | 0.502±0.027 | 0.079±0.046 | -0.002±0.037 |
| RETCLIP | 0.538±0.087 | 0.547±0.033 | 0.130±0.040 | 0.051±0.037 |
| PMC-CLIP | 0.484±0.048 | 0.474±0.030 | 0.054±0.039 | -0.031±0.033 |
| RETFound+GatorTron | 0.640±0.062 | 0.577±0.067 | 0.183±0.095 | 0.121±0.101 |
| Ours (no GACL) | 0.653±0.072 | 0.596±0.070 | 0.187±0.092 | 0.135±0.096 |
| Ours (with GACL) | **0.659±0.073** | **0.605±0.070** | **0.189±0.091** | **0.140±0.096** |

and MCC, but at the cost of a reduced AUROC. This highlights the importance of carefully calibrated thresholds, as multimodal associations are highly sensitive to pairing phenotypically similar pairs and avoiding weakly related alignments. Distinct trends were observed between image and text modalities. For images, higher thresholds demonstrated better performance, suggesting that lower thresholds introduce noise by forcing dissimilar samples to be similar. Conversely, for text embeddings, lower thresholds led to higher predictive performance, indicating that learning benefits when a broader range of semantically related texts are considered similar. In addition, the observed trade-off between accuracy, F1-score, MCC, and AUROC at higher image thresholds in incident AD prediction reflects a point estimate classification performance and ranking-based discrimination in prediction performance analysis using AUROC. In the incident AD prediction task, a higher image threshold forced the stricter alignment, which improved classification performance at a fixed operating point. However, the ranking ability across different thresholds was reduced as a tradeoff, leading to a lower accuracy. Therefore, the generalizability of these trends requires further validation in other domains and different datasets to validate the threshold-dependent trade-off influenced by dataset-specific factors.

### 3.2.3. IMPACT OF CLINICAL VS. LATENT SIMILARITY IN GACL

The AD and dementia prediction results using morphometric features and the model's latent features in image-image similarity computation of GACL are shown in Table 4. For this experiment, the threshold for the image latent was determined as the third quartile of the similarity distribution in the development set ($\tau_F$=0.9974). In both the incident AD and dementia prediction cases, incorporating morphometric features consistently yielded superior performance. This indicates that clinically grounded morphometric similarity provides a more reliable and meaningful signal for identifying individuals who share similar retinal and systemic phenotypes, enabling richer and more discriminative representational learning.

### 3.2.4. IMPACT OF DIFFERENT LOGICAL OPERATORS IN GACL

The comparative analysis result between logical OR and AND operators in GACL for both incident AD and dementia prediction task are shown in Table 11 (Appendix J). Across

both tasks, the OR and AND operators yielded nearly identical AUROC values, indicating comparable performance across different thresholds of the SVM classifier. However, the OR operator consistently achieved higher balanced accuracy, F1-Score, and MCC compared to the AND operator. This trend indicates that the requirement for similarity from at least one modality is more effective than a stricter similarity criterion. Thus, the OR operator provides greater flexibility by capturing partially overlapping phenotypic signals, leading to improved classification performance.

Table 4: Performance of the incident AD and dementia prediction with different image-image similarity methods

|  | AUROC | Balanced Accuracy | F1-Score | MCC |
|---|---|---|---|---|
| **AD** |  |  |  |  |
| Latent Feature | 0.656±0.062 | 0.592±0.079 | 0.201±0.105 | 0.140±0.111 |
| Morphometric Feature | **0.658±0.095** | **0.610±0.083** | **0.208±0.105** | **0.147±0.117** |
| **Dementia** |  |  |  |  |
| Latent Feature | 0.654±0.055 | 0.594±0.052 | 0.181±0.067 | 0.134±0.075 |
| Morphometric Feature | **0.659±0.073** | **0.605±0.070** | **0.189±0.091** | **0.140±0.096** |

## 4. Conclusion

In this paper, we present REVEAL, a multimodal VLM framework that improves embedding learning for incident AD and dementia prediction by explicitly aligning retinal morphometric features with individualized risk factors. Our group-aware contrastive learning strategy identifies clinically meaningful groups and patients with similar retinal and risk profiles, and enhances cross-modal representation learning. This alignment improves AD and dementia prediction diagnosed after an average of 8 years after the baseline visit. These gains demonstrate that multimodal alignment reflects the strong correspondence between AD-specific risk factors and retinal structural features. Moreover, transforming structured clinical data into narrative form leverages the semantic richness of pretrained language models, further strengthening multimodal associations and boosting predictive performance. These results underscore the value of clinically contextualized representation learning in VLMs for early AD and dementia risk stratification. Despite promising results, several limitations should be acknowledged. First, the performance of the REVEAL is sensitive to the threshold selection in GACL, reflecting a trade-off between strict phenotypic alignment and preserving sufficient shared representation for robust learning. Second, our evaluation is limited to a single large cohort (UK Biobank) with a limited number of incident cases of AD and dementia, limiting the generalizability of REVEAL to other populations and other disease settings. Finally, the evaluation of prompt variants for better alignment performance should be further evaluated. While absolute predictive performance remains limited by cohort size and disease prevalence, the consistent relative gains demonstrate the value of clinically grounded multimodal alignment for long-horizon neurodegenerative risk modeling.

## Acknowledgments

This research has been conducted using data from UK Biobank, a major biomedical database under application ID 48388. This material is based upon work supported by the National Science Foundation under Grant No. (NSF 2123809).

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

## Appendix A. Full template for clinical report generation

Template: The subject is <age> years old <ethnic background> <sex>. The average total household of this subject is in between <economic status>. The subject has <HbA1C> HbA1C, <HDL> HDL, <BMI> BMI, <systolic blood pressure> systolic blood pressure,

<diastolic blood pressure> diastolic blood pressure. For lifestyle, the subject is in <employment status>. The subject is <smoking history>, has <depression>, has sleep deprivation <sleep deprivation>, and drinks alcohol <alcohol use>. The subject had his first cannabis at age <age of cannabis initiation> and used cannabis <cannabis use> times. The subject visits family <frequency of family visit>, and <number of leisure activity>. For physical activity, the subject walks <duration of walked 10+ minutes> minutes <number of days/week of walked 10+ minutes> days per week, exercises moderately <duration of moderate activity> minutes for <number of days/week of moderate activity> days a week, and exercises vigorously <duration of vigorous exercise> minutes for <number of days/week of vigorous activity> days a week. For diet, the subject has <cooked vegetable intake> tablespoons of cooked vegetables, <raw vegetable intake> tablespoons of raw vegetables, <fresh fruit intake> tablespoons of fresh fruit, and <dried fruit intake> dried fruit. In addition, the subject has oily fish <oily fish intake>, non-oily fish <non oily fish intake>, processed meat <processed meat intake>, poultry <poultry intake>, beef <beef intake>, lamb <lamb intake>, and pork <pork intake>. The subject has <bread intake> slices of bread per week, with <spread type>. The subject drinks <milk type>, <tea intake> cups of tea, <coffee intake> cups of coffee, <water intake> cups of water per day. The subject puts <salt added to food> in his diet. For cognitive function, the subject remembered <numeric memory> digits in the numeric memory test, scored <fluid intelligence> in a fluid intelligence test, completed trail #1 in <trail-making test A duration> deciseconds with <trail-making test A error counts> errors, and completed trail #2 in <trail-making test B duration> deciseconds with <trail-making test B error counts> errors.

When a risk factor was unavailable (e.g., age of cannabis initiation), the report stated: **No cannabis use was reported at that age** in the <age of cannabis initiation> section.

## Appendix B. Implementation details and hyperparameter discovery

The dimension of the projection layer for both image and text encoders was fixed at 1024. The batch size was fixed at 128. The parameter search space and determined values for REVEAL are available in Table 5. The ranges for $\tau_F$ and $\tau_T$ were determined by the 3rd quartile to the 4th quartile range of retinal morphometric similarities and pseudo-clinical report similarity in 85% of the development set. Based on Optuna, learning rate was determined as 2.42e-4, eps was determined as 8.61e-7, weight decay was set to 0.0232, thresholds were determined as $\tau_F$=0.9481 and $\tau_T$=0.9808. When training without GACL, we used the standard InfoNCE loss.

Table 5: Hyperparameter search space and optimal values

| Hyperparameter | Range (min, max) | Optimal Value |
| --- | --- | --- |
| learning rate | 1e-6, 5e-4 | 2.42e-4 |
| eps | 1e-9, 1e-6 | 8.61e-7 |
| weight decay | 1e-6, 1e-1 | 0.0232 |
| $\tau_F$ | 0.2853, 0.9949 | 0.9480 |
| $\tau_T$ | 0.9548, 0.9979 | 0.9808 |
| $\beta$ | -5, 0 | -0.6319 |

## Appendix C.  Demographic information of incident AD and dementia subjects and controls

Table 6: SVM train-test splits and demographic characteristics of subjects with incident Alzheimer's Disease (AD) and controls

|  | With incident AD (n=86) | Without incident AD (n=1077) |
|---|---|---|
| $SVM_{train}/SVM_{test}$ | 69/17 | 862/215 |
| Gender: # male (%) | 45 (52.33) | 550 (51.07) |
| Age: mean (s.d) | 64.23 (3.81) | 64.31 (3.73) |
| Ethnicity: caucasian % | 86.05 | 97.55 |

Table 7: SVM train-test splits and demographic characteristics of subjects with incident dementia and controls

|  | With incident dementia (n=93) | Without incident dementia (n=1139) |
|---|---|---|
| $SVM_{train}/SVM_{test}$ | 74/19 | 911/228 |
| Gender: # male (%) | 50 (53.76) | 607 (53.29) |
| Age: mean (s.d) | 64.54 (3.87) | 64.24 (3.84) |
| Ethnicity: caucasian % | 86.02 | 97.28 |

## Appendix D.  Distribution of disease onset of Alzheimer's Disease and Dementia

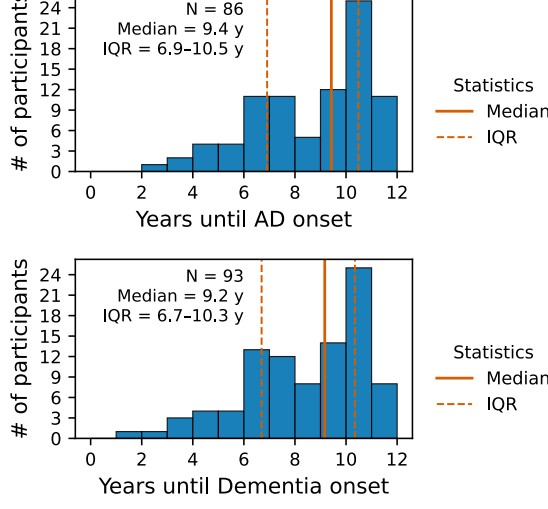

Figure 4: The years until onset of Alzheimer's Disease and dementia. IQR denotes interquartile range.

## Appendix E. Full list of AD and dementia risk factors used in this study

- Demographic Information ($d = 5$): Age, sex, economic status, ethnic background, employment status

- General Health Information ($d = 11$): BMI, HbA1C, HDL, systolic/diastolic blood pressure, numeric memory, fluid intelligence, Trail-Making Test A/B duration and error counts

- Risk Factors ($d = 6$): Depression, sleep deprivation, alcohol use, smoking history, cannabis use, age of cannabis initiation

- Physical activity ($d = 6$): Number and Duration of days/week walked 10+ minutes, Number and Duration of days/week of moderate physical activity 10+ minutes, Number and Duration of days/week of vigorous physical activity 10+ minutes

- Social and leisure activities ($d = 2$): Frequency of friend&family visit, number of leisure activity

- Dietary habits ($d = 18$): cooked vegetable intake, raw vegetable intake, fresh fruit intake, dried fruit intake, oily fish intake, non-oily fish intake, processed meat intake, poultry intake, beef intake, lamb intake, pork intake, milk type, spread type, bread intake, salt added to food, tea intake, coffee intake, water intake

## Appendix F. Full list of fundus-based retinal morphometry used in this study

- Optic nerve head features($k = 2$): Vertical and horizontal cup-to-disc ratios.

- Vascular features ($k = 15$): Fractal dimension, fractal density, distance tortuosity, squared curvature tortuosity, and tortuosity density for artery, vein, and both combined.

## Appendix G. Statistical comparison of REVEAL with baseline and other multimodal methods for incident AD and Dementia prediction

Table 8: Welch's t-test results and Hedges' g effect sizes for model performance in incident AD prediction. Each cell reports the p-value and corresponding effect size. See Table 2 for absolute performance values

|  | AUROC | Balanced Accuracy | F1-Score | MCC |
|---|---|---|---|---|
| Baseline SVM | 0.09 (0.75) | 0.35 (0.41) | 0.13 (0.67) | 0.16 (0.63) |
| KeepFIT-CFP | 0.00 (1.87) | 0.01 (1.32) | 0.03 (1.09) | 0.00 (1.39) |
| BiomedCLIP | 0.00 (1.56) | 0.01 (1.21) | 0.04 (0.99) | 0.01 (1.29) |
| RETCLIP | 0.02 (1.11) | 0.01 (1.20) | 0.02 (1.09) | 0.01 (1.26) |
| PMC-CLIP | 0.00 (2.35) | 0.00 (1.98) | 0.00 (1.66) | 0.00 (1.92) |
| RETFound+GatorTron | 0.92 (0.04) | 0.26 (0.49) | 0.48 (0.31) | 0.42 (0.35) |

Table 9: Welch's t-test results and Hedges' g effect sizes for model performance in incident dementia prediction. Each cell reports the p-value and corresponding effect size. See Table 3 for absolute performance values

|  | AUROC | Balanced Accuracy | F1-Score | MCC |
|---|---|---|---|---|
| Baseline SVM | 0.03 (1.01) | 0.22 (0.54) | 0.26 (0.50) | 0.08 (0.83) |
| KeepFIT-CFP | 0.00 (2.82) | 0.00 (1.16) | 0.03 (1.09) | 0.00 (1.76) |
| BiomedCLIP | 0.00 (2.73) | 0.00 (1.86) | 0.00 (1.45) | 0.00 (1.86) |
| RETCLIP | 0.00 (1.43) | 0.03 (1.00) | 0.09 (0.80) | 0.02 (1.16) |
| PMC-CLIP | 0.00 (2.70) | 0.00 (2.33) | 0.00 (1.84) | 0.00 (2.27) |
| RETFound+GatorTron | 0.53 (0.27) | 0.38 (0.38) | 0.89 (0.05) | 0.68 (0.18) |

## Appendix H. Component-wise ablation results for REVEAL

Table 10: Component-wise ablation results for REVEAL on incident Ad and dementia prediction. Image-only uses image embeddings alone; Image+Table combines image embeddings with raw tabular risk factors; Text-only uses LLM-derived clinical narrative embeddings; and Image+Text jointly models image and text embeddings. Model's performance is reported as mean±standard deviation across 10 runs

|  | AUROC | Balanced Accuracy | F1-Score | MCC |
|---|---|---|---|---|
| **AD** |  |  |  |  |
| Image-only | 0.561±0.056 | 0.527±0.039 | 0.117±0.044 | 0.029±0.044 |
| Image+Table | 0.587±0.077 | 0.559±0.075 | 0.131±0.086 | 0.061±0.091 |
| Text-only | 0.630±0.074 | 0.573±0.057 | 0.188±0.099 | 0.111±0.105 |
| Image+Text | 0.658±0.095 | 0.610±0.083 | 0.208±0.105 | 0.147±0.117 |
| **Dementia** |  |  |  |  |
| Image-only | 0.518±0.050 | 0.523±0.037 | 0.116±0.043 | 0.089±0.030 |
| Image+Table | 0.559±0.083 | 0.553±0.056 | 0.134±0.063 | 0.056±0.065 |
| Text-only | 0.641±0.042 | 0.583±0.059 | 0.168±0.076 | 0.105±0.086 |
| Image+Text | 0.659±0.073 | 0.605±0.070 | 0.189±0.091 | 0.140±0.096 |

## Appendix I. Impact of Thresholds on REVEAL Performance

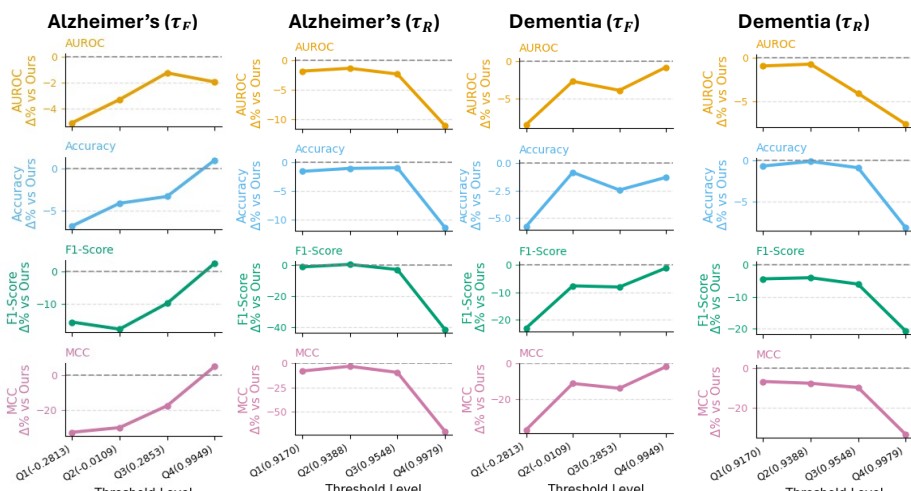

Figure 5: Effect (% difference) of varying thresholds on the incident AD and dementia prediction task.

## Appendix J. Performance of REVEAL with OR and AND operation

Table 11 compares logical OR and AND operations in the GACL. While both strategies yield comparable AUROC, the OR operation consistently achieves equal or slightly better Balanced Accuracy, F1-score, and MCC across both tasks, indicating that enforcing similarity in either modality is more effective than requiring simultaneous agreement in both.

Table 11: Performance Comparison between OR and AND function in GACL

|  | AUROC | Balanced Accuracy | F1-Score | MCC |
|---|---|---|---|---|
| **AD** |  |  |  |  |
| AND | 0.659±0.094 | 0.607±0.082 | 0.205±0.103 | 0.144±0.115 |
| OR | 0.658±0.095 | 0.610±0.083 | 0.208±0.105 | 0.147±0.117 |
| **Dementia** |  |  |  |  |
| AND | 0.659±0.075 | 0.602±0.071 | 0.184±0.090 | 0.135±0.095 |
| OR | 0.659±0.073 | 0.605±0.070 | 0.189±0.091 | 0.140±0.096 |

