# OpenReview forum: "REVEAL: Multimodal Vision–Language Alignment of Retinal Morphometry and Clinical Risks for Incident AD and Dementia Prediction"
_MIDL.io/2026/Conference — MIDL 2026 Poster_

### Official Review · Reviewer_5jtu · 2025-12-30

**Confidence:** 3
**Preliminary Rating:** 4
**Final Rating:** 4

**Summary:**

The paper introduces a multimodal framework designed for the early prediction of incident AD and dementia. The authors address the challenge of integrating structured clinical risk factor data with retinal imaging by using a LLM to convert tabular questionnaires into clinically descriptive narratives. To improve the alignment between these modalities, they propose a Group-Aware Contrastive Learning strategy that identifies and clusters phenotypically similar subjects based on retinal morphometry and risk profiles, moving beyond traditional one-to-one sample pairing.

**Strengths:**

1. GACL is a well-motivated departure from standard CLIP-style 1:1 pairing. By using explicit clinical grounding (retinal morphometric features) to define positive pairs within a batch, the model effectively captures shared pathophysiological patterns across different subjects.
2. The use of LLMs to synthesize standardized clinical reports from structured data is a highly practical solution to the modality gap.
3. The authors compare their method against a wide array of strong baselines, including specialized medical VLMs (BiomedCLIP, PMC-CLIP) and fundus-specific foundation models (RETFound), showing consistent improvements across AUROC, MCC, and F1-score.

**Weaknesses:**

1.  The authors state that all participants who later developed incident AD/dementia were assigned to the test set, while the training/validation sets consisted only of healthy controls. While this preserves "preclinical" integrity, it means the model's representation alignment is learned entirely on healthy eyes. It is unclear if excluding the "to-be-diseased" samples from Stage 1 prevents the model from learning specific early-stage structural-clinical correlations that might only exist in the incident group.
2. The final prediction is performed using an SVM on top of frozen embeddings. While this demonstrates the quality of the learned representations, an end-to-end fine-tuning comparison or a more modern MLP-based classifier could have been explored to see if further gains are possible.
3.  While the authors mention using a CARE-based template to ensure consistency, there is limited discussion on how sensitive the final performance is to the specific wording or structure of the LLM-generated clinical narratives.

**Detailed Comments:**

In Figure 3, the "Indicator Function" notation could be slightly larger for better readability.

**Justification Of Final Rating:**

Regarding the exclusion of incident cases during Stage-1 alignment, the authors provide a clear and reasonable justification grounded in leakage prevention and data availability.  While I still believe that selectively incorporating incident cases could be an interesting extension, the current design is coherent and well-motivated, especially given the limited number of cases after QC.

On the use of a frozen encoder with an SVM classifier, the authors’ explanation that this choice isolates representation quality is satisfactory. However, I think an end-to-end MLP-based comparison could potentially yield higher absolute performance.

**Justification Of The Preliminary Rating:**

The paper presents a new framework to solve a common data-type mismatch in medical imaging. The results are compelling, particularly the performance on a challenging longitudinal prediction task (8 years lead time).

**Questions To Address In The Rebuttal:**

1. Why were incident cases strictly excluded from the training/alignment phase? Would including a portion of these cases in training allow the model to learn more disease-specific multimodal clusters during GACL?
2. Regarding the LLM narratives: Did the authors experiment with multiple prompts or "chain-of-thought" style summaries to see if the level of medical abstraction in the text affects the alignment quality?
3. How does the model perform if only one modality is used during the inference/prediction stage? This would help clarify if the alignment truly "enriched" the image embeddings or if the performance is primarily driven by the text embeddings in the SVM.
4. Were the images used in Stage 1 and Stage 2 from the same visit? (The text implies baseline visit, but confirmation would be helpful).

---

> ### Author Response · Authors · 2026-01-23
> **Rebuttal Response to Reviewer 5jtu**
>
> We thank the reviewer for a positive and constructive evaluation of our work. We appreciate the reviewer’s recognition of the novelty and clinical motivation of the proposed Group-Aware Contrastive Learning (GACL) framework, as well as the practical use of LLM-generated clinical narratives. Below, we address each of the raised questions and concerns in detail.
>
> **1. Exclusion of incident cases during Stage-1 alignment:** Incident AD and dementia cases were excluded from the Stage-1 alignment phase to maintain a strictly preclinical, leakage-free training regime, ensuring that representation learning is not influenced by the future outcomes. We designed Stage-1 to capture normative retinal-risk relationships, such that predictive signals in Stage-2 purely arise from subtle baseline deviations rather than disease-specific patterns. This was also motivated by the data availability: within UKB, after fundus image quality control, only 107 prevalent and incident all-cause dementia cases remained, limiting their effective use in alignment. However, we agree that incorporating a subset of incident cases could facilitate disease-specific multimodal grouping and lead to better prediction results. We would approach the reviewer’s suggestion as future work.
>
> **2. End-to-end fine-tuning comparison or a more modern MLP-based classifier:** A stronger classifier would confound attribution of gains, whereas a simple SVM provides a conservative estimate of representation quality. However, we also agree with the reviewer’s opinion that end-to-end fine-tuning or a modern MLP-based classifier might achieve further gain. Exploring end-to-end fine-tuning or MLP-based classifiers is a promising direction for future work and may further improve absolute performance.
>
> **3. Question Regarding LLM Narratives:** In this study, we did not experiment with multiple prompts or “chain-of-thought style summaries” to enhance multimodal alignment with richer language. Our goal was to serialize the structured questionnaire variables into a clinically plausible text format that a medical text encoder can reliably ingest. To this end, we employed a fixed, CARE-inspired template with deterministic 1:1 mapping from tabular to text to explicitly handle the missing values and avoid stylistic variations. This was intended to minimize the sensitivity to prompt choice and to avoid introducing an additional degree of freedom for stability and auditability. We did not further explore the multiple prompts or chain-of-thought style summaries in this work, as such approaches are known to introduce variability and potential hallucination. However, we agree with the reviewer that text abstraction level and prompt variants could affect the alignment performance (doi s41746-025-01433-4, s41598-025-26705-7). As future work, we can systematically evaluate prompt variants with different levels of abstraction to explicitly compare prompting strategies. We explicitly revised the Conclusion on Page 12 to state this as a limitation of our work.
>
> **4. One modality for prediction:** Section 3.2.1’s 2nd paragraph and Table 10 in Appendix H include the component-wise ablation test results to analyze each image and text embedding component’s contribution in incident prediction. While Text-only primarily drives the prediction with rich medical semantics, enriched image representation complements the information that text representation alone cannot convey.
>
> **5. Clarification of images used in Stage 1 and Stage 2:** Yes, all images and risk factor data used in this study came from the baseline assessment visit.

---

### Official Review · Reviewer_iPeB · 2026-01-08

**Confidence:** 3
**Preliminary Rating:** 4
**Final Rating:** 4

**Summary:**

In this article, they tackle the fascinating task of predicting preclinical Alzheimer's disease (AD) and dementia using color fundus photography (CFP) and disease-specific risk profiles from the UK Biobank. To integrate disease-specific risk profiles, they jointly model retinal biomarkers and clinical risk factors by introducing group-aware contrastive learning (GACL) to link clinical reports with CFP. They outperform both state-of-the-art retinal imaging models combined with clinical text encoders and general image-language models (VLMs).

**Strengths:**

- The authors tackle the challenging task of integrating clinical risk factors into a vision-language model, as direct application to CLIP embeddings is not possible. They solve this technical challenge by synthesizing standardized clinical-style narratives from tabular health data, converting each participant's risk factor profile into a synthetic clinical report.
- Their main contribution is innovative: the clinical alignment of individuals with shared morphometric characteristics of the retina *or* with the same clinical risk factors using their GACL method.
- The performance improvements achieved by GACL are convincing.
- The comparison of the morphometric features to the CFP encoder embeddings in Table 4 shows a clear advantage of the predefined morphometric features.

**Weaknesses:**

- It is clear that a SVM is trained for the binary classification of preclinical AD and dementia. But how exactly does one arrive at the average performance of 8 years before diagnosis (range 1-11 years)? Is this determined based on the time of acquisition of the images under consideration and when the disease was first diagnosed?
- Please report the standard deviations between the 10 random seeds of each downstream classification model.
- Please use statistical tests to further emphasize that the proposed approach significantly outperforms the other models.
- Tables 2 + 3: What is the difference between "RETFound+
GratorTron" and "Ours (no GACL)"? Is "Ours (no GACL)" only trained with positive pairs on the main diagonal of the image-text similarity matrix (as with a standard CLIP embedding), and is "RETFound+GratorTron" merely a concatenation of both embeddings?

**Detailed Comments:**

- Figure 1: The term "image embedding" is confusing here, as it refers to the embedding of predefined features extracted from fundus images, which are also referred to as "morphometric features" in the paper. In addition, the authors later also refer to an "image embedding" generated by a fundus image encoder.
- Figure 1: "Ground Truth Matrix for Contrastive Learning" does not make sense yet. I recommend renaming it "Intra-Modality Pair Matrix".
- Figure 2: Please remove the red underlining from the text if it has no meaning.
- Do you have any analyses or findings on the use of the logical AND operator for the group-aware contrastive learning strategy - i.e., that both the morphology and the clinical findings must be similar (up to a certain threshold)? Or would this be too strict a criterion?
- The first sentence in section 3.1.2 is confusing. Aren't all models except RETFound multimodal vision-language models?

**Justification Of Final Rating:**

I appreciate that the authors addressed all of my points during the rebuttal, and I find the revised version of the paper clearer and stronger. I also thank the authors for responding to my concerns regarding the statistical tests used for k-fold cross-validation. However, after considering the paper in its entirety, I will maintain my original rating.

**Justification Of The Preliminary Rating:**

The article presents an innovative and effective approach to multimodal image-language alignment for predicting preclinical AD and dementia based on CFP and clinical reports. However, standard deviations and statistical tests are not provided in the results tables.

**Questions To Address In The Rebuttal:**

Please address my points in the sections "Weaknesses" and "Detailed Comments".

---

> ### Author Response · Authors · 2026-01-23
> **Rebuttal Response to Reviewer iPeB**
>
> We thank the reviewer for acknowledging the difficulty of the task we are tackling in this study and the innovative aspect of our method. We have addressed all weaknesses and comments in our revised version, and here is the one-by-one reply.
>
> **We have revised the figures and text accordingly:** (1) Figure 1: We revised the figure 1’s arrow to make the concept less confusing. However, the image embedding and morphometric features are distinctly different. Image embedding is an encoded embedding directly from the image encoder models, and morphometric features are 17 different features that quantitatively define retinal anatomy, which is used in computing fundus similarity in GACL. The “Ground Truth Matrix for Contrastive Learning” was replaced with “Intra-Modality Pair Matrix for Contrastive Learning”. (2) Figure 2: The red underlining was removed.
>
> **1. Clarification of 8 years before diagnosis:** We apologize for the confusion. The lead time of “N years before diagnosis” was acquired by the difference between the baseline CFP acquisition date (ID:53) and the first recorded AD (or dementia) diagnosis date in the UK Biobank fields (ID: 42018, 42020, 42022, and 42024). In SVM, the positive case of AD (or dementia) was considered as 1, and all controls were considered as 0 for a binary label. We clarified this explicitly on Page 8, Section 3.1.1.
>
> **2. Standard Deviations & Statistical Tests:** We thank the reviewer for this suggestion. All experiments were repeated across 10 random seeds. We included mean ± standard deviation in Tables 2 and 3, and included statistical testing (e.g., Welch’s t-test and Hedge’s g in Tables 8 and 9 in Appendix D) in the revised manuscript. We note that because each comparison is based on 10 runs, the associated hypothesis tests are statistically underpowered to detect small-to-moderate effect size (e.g., at p-value of 0.05, detectable effect sizes are necessarily large, 0.9 in our case). As a result, non-significant p-values should not be interpreted as evidence of equivalence. We therefore emphasize effect sizes and the consistency of performance improvements across tasks and metrics as the primary indicators of model performance differences.
>
> **3. RETFound+GatorTron vs Ours (no GACL):** We are sorry for the confusing notation. As the reviewer pointed out, “RETFound+GatorTron” refers to a concatenation of independently pretrained image and text embeddings without contrastive alignment, and “Ours (no GACL)” uses CLIP-style contrastive alignment between fundus images and LLM-generated clinical narratives but restricts positives to the same subject pairs (main diagonal of the image-text similarity matrix) only. We included an additional explanation of RETFound+GatorTron in Section 3.1.2 on Page 8.
>
> **4. OR vs AND in Group Similarity Computation:** We included the additional experiment comparing the OR and AND grouping in GACL (Sections 3.1.5, 3.2.4, Table 11 in Appendix J). As shown in Table 11 from Appendix F, forcing the similarity in at least one modality. However, we argue that our original intention to use a logical OR rather than AND was to avoid overly restrictive grouping. The structural change in the retina can occur due to different risk conditions with comorbidities, and some health conditions precede detectable changes in retinal anatomy. AND operation will exclude some of the clinically meaningful pairs, leading to a less effective representation learning.
>
> **5. Minor revision:** Section 3.1.2 (Page 8) was revised to clarify that RETFound is image-only, while other models are multimodal.

---

> > ### Comment · Reviewer_iPeB · 2026-01-28
> >
> > I thank the authors for addressing my comments in detail. However, a concern regarding the validity of the statistical tests remains. While the authors appropriately caution against overinterpreting non-significant p-values due to limited power, this does not resolve the more fundamental issue that Welch’s t-test assumes independent samples -- an assumption violated by k-fold cross-validation because of overlapping training sets. As a result, the reported p-values may be biased regardless of power considerations. Emphasizing effect sizes (e.g., Hedge’s g) and consistency across tasks is appropriate and informative; however, if hypothesis testing is retained, the authors should employ statistical tests specifically designed for cross-validated model comparisons (e.g., the corrected resampled t-test or the Friedman test with appropriate post-hoc analysis).
> >
> > For clarification, are all comparisons made between a baseline and the proposed model as a fixed reference? If so, then larger Hedge’s g values indicate that the proposed model outperforms the baseline, correct?

---

> > > ### Author Response · Authors · 2026-01-28
> > > **Reply to Additional Comment from Reviewer iPeB**
> > >
> > > Thank you for highlighting the important concern regarding the statistical testing of our results. All comparisons in the statistical test are performed using the proposed model as a fixed reference. Thus, larger values of Hedges' g indicate superior performance of the proposed model relative to the baseline.
> > >
> > > We also acknowledge that Welch's t-test relies on an independence assumption that is violated under cross-validation. In response to this concern, we employed statistical analysis to rely on tests that are appropriate for cross-validated model comparisons. Specifically, we employ a Friedman test across methods, followed by a paired post-hoc Wilcoxon signed-rank test comparing the fixed reference to each baseline, with correction for multiple comparisons. Friedman test results were all significant, with a post-hoc test as a complementary result.
> > >
> > > Consistent with the reviewer’s recommendation, we place primary emphasis on effect sizes and on the consistency of performance across tasks and metrics, and we treat hypothesis testing as a complementary analysis rather than the basis of our conclusions. We will incorporate these clarifications and results in the final version.

---

### Official Review · Reviewer_cUWa · 2026-01-12

**Confidence:** 5
**Preliminary Rating:** 4
**Final Rating:** 4

**Summary:**

The paper introduces, a multimodal framework designed to predict incident Alzheimer’s Disease (AD) and dementia an average of 8 years before clinical onset. The authors address the "modality gap" between structured tabular risk factors and retinal images by using a LLM to synthesize clinical narratives. To improve alignment, they propose Group-Aware Contrastive Learning (GACL), which identifies and clusters clinically similar subjects as positive pairs during training based on their retinal morphometry and risk profiles. The model is validated on the UK Biobank dataset, demonstrating superior performance over retinal-only models and general vision-language models.

**Strengths:**

1.	GACL uses explicit retinal morphometric features to identify phenomenologically similar individuals. This provides a much-needed clinical anchor for contrastive learning in medical imaging.
2.	Using an LLM to convert structured questionnaire data into standardized clinical narratives allows the framework to leverage powerful pretrained vision-language models (VLMs) that were originally trained on natural language

**Weaknesses:**

1. Despite outperforming baselines, the absolute performance metrics remain concerningly low for clinical utility. For incident AD, the F1-Score is only 0.2270 and the MCC is 0.1562. These values suggest a high rate of false positives or negatives, which limits the model's current use as a standalone screening tool.

2. The number of incident AD cases (86) is very small compared to the control pool (1,077). While class-weighted training was used, this scarcity likely contributes to the low F1 scores and may lead to overfitting on the specific characteristics of those few cases.

3. The cohort is derived from the UK Biobank. This raises significant questions regarding the generalizability of the retinal biomarkers and lifestyle risk factors to more diverse global populations.

**Detailed Comments:**

A fundamental conceptual concern regarding the REVEAL framework is the nature of the "alignment" being learned between retinal images and clinical risk factors. In traditional vision-language models (VLMs) like CLIP, pre-training relies on explicit visual-semantic mapping, where the image contains the literal objects described in the text. However, in REVEAL, there is no direct visual representation of "smoking history" or "economic status" within a fundus photograph. What is the pretraining exactly learning?

In constructing the group similarity mask, the authors use a logical OR between the image similarity mask and the text similarity mask. This means subjects are grouped if they are similar in either retinal structure or clinical risk. Why don’t use logical AND, which might produce cleaner clusters?

Figure 5 reveals that the model is highly sensitive to the choice of threshold. The trend shows that higher image thresholds generally improve accuracy but decrease AUROC for AD. It is suggested to provide more justification.

A major experimental gap is the lack of a baseline comparing REVEAL to a direct fusion of RETFound image latents and raw tabular features. Without this, it is unclear if the improvement stems from the "semantic richness" of the LLM narrative or simply the power of the image foundation model.

**Justification Of Final Rating:**

I have decided to maintain my rating of Weak Accept. The authors provided a comprehensive rebuttal that effectively addressed the primary technical and methodological concerns raised during the review process.

**Justification Of The Preliminary Rating:**

The paper introduces REVEAL, a novel framework for predicting incident Alzheimer's Disease and dementia an average of 8 years before diagnosis by aligning retinal morphometry with clinical risk factors. The technical approach of using LLMs to synthesize clinical narratives from tabular data and the proposed group-aware contrastive learning strategy are innovative methods to bridge the modality gap. However, the absolute performance metrics remain concerningly low for clinical utility, with an F1-Score of only 0.2270 and an MCC of 0.1562 for incident AD prediction. Finally, the conceptual nature of aligning non-visual factors with fundus images and the use of a logical OR for grouping subjects require deeper theoretical and experimental justification.

**Questions To Address In The Rebuttal:**

Why was a logical OR used for the group similarity mask instead of a logical AND? Could an "AND" operation produce cleaner, more clinically relevant clusters by requiring similarity in both modalities?

In contrast to traditional VLMs like CLIP that use explicit visual-semantic mapping, REVEAL aligns images with factors like "smoking history" that have no direct visual presence. What exactly is the model learning during this pre-training, and how do you ensure it is capturing pathophysiological correlations rather than noise?

To isolate the "semantic richness" of the LLM narrative, will you provide a baseline comparing REVEAL to a direct fusion of RETFound image latents and raw tabular features?

---

> ### Author Response · Authors · 2026-01-23
> **Rebuttal Response to Reviewer cUWa**
>
> We thank the reviewer for acknowledging the novelty of the REVEAL framework for incident Alzheimer’s disease (AD) and dementia prediction and for highlighting important points to improve clarity. We have addressed all comments in the revised manuscript as follows.
>
> **1. Low absolute performance for clinical utility:** We agree that the absolute performance does not yet support standalone clinical screening. This is expected given the extreme difficulty of the task: predicting incident AD and dementia on average 8.7 years before diagnosis using only baseline, non-invasive retinal images and questionnaire-derived risk factors, without disease-specific biomarkers. Our goal is methodological rather than immediate deployment. Compared to strong baselines, including RETFound + GatorTron, PMC-CLIP, and tabular SVMs, REVEAL achieves consistent relative improvements across metrics, suggesting that multimodal alignment captures preclinical signals not accessible to prior models.
>
> **2. The limited number of incident AD cases (86) and the control pool:** We acknowledge that the number of incident AD and dementia cases is limited, which reflects the challenge of preclinical prediction in population scale cohort. In UKB, after fundus image quality control, only 107 prevalent and incident all-cause dementia cases remained. Importantly, REVEAL is evaluated under a strictly hard setting where modest F1 and MCC values were expected and consistent with a prior study (PMID: 21498903). We view the current result as evidence of meaningful signal extraction despite severe data constraints. Therefore, a larger cohort will be essential to further improve performance stability and clinical utility.
>
> **3. Generalizability Beyond UK Biobank:** We agree that generalizability is limited by the UK Biobank cohort. While UKB is not globally representative, our primary goal was to demonstrate the association between retinal morphometry and epidemiological risk factors. We have revised the discussion to explicitly acknowledge this limitation and emphasize the need for validation in ethnically and geographically diverse cohorts (Conclusion, Page 12).
>
> **4. What is being aligned?:** REVEAL does not assume a direct visual-semantic correspondence (e.g., objects & words) like natural image CLIP. Instead, it learns the pathophysiological relationship between retinal structure and systemic risk factors. Risk factors such as smoking, hypertension, or depression are not visually observable, but they are known to induce microvascular and optic nerve head changes in the retina. The contrastive objective, therefore, encourages alignment between retinal structural patterns and clinically meaningful risk profiles, enabling the model to learn a latent disease-risk manifold rather than literal visual semantics. Noise is mitigated through explicit clinical grounding using retinal morphometry-based similarity rather than latent image embeddings. If the alignment were dominated by noise, we would expect no improvement over RETFound+GatorTron or degradation when introducing GACL, which is not observed. We clarified this distinction in the revised manuscript (Section 2.2.2, Page 4).
>
> **5. OR vs AND in group similarity computation:** We adopted a logical OR rather than AND to avoid overly restrictive grouping. In preclinical disease, similarity may manifest asynchronously across modalities: individuals may share high-risk clinical profiles before detectable retinal changes, or vice versa. An AND operation would exclude such clinically meaningful pairs and reduce the learning signal. To validate this choice, we added experiments showing that OR consistently achieves higher balanced accuracy, F1-score, and MCC across both prediction tasks (Sections 3.1.5, 3.2.4; Table 11, Appendix J).
>
> **6. Threshold-related justification:** We thank the reviewer for highlighting the sensitivity of REVEAL to the threshold choice. In the revised manuscript, we have added a detailed justification (Page 11, Section 3.2.2) to explain the observed trade-off. Specifically, a higher threshold enforced stricter alignment criteria, improving the performance at a fixed operating (threshold for classification) point. However, the trade-off of rank-based discrimination ability across different operating points led to lower AUROC. This clarification highlights the importance of carefully calibrated thresholds in multimodal contrastive learning.
>
> **7. Missing baseline:** We agree that comparing the direct fusion of REVEAL’s image embeddings and tabular risk factors clarifies the source of performance gains. As shown in Table 10 (Appendix F), Image+Table is outperformed by both Text-only and Image+Text, indicating that the pseudo-clinical narrative provides rich semantic information. Image-only and Text-only results further show that, while the clinical narrative is the primary driver, image and text embeddings are complementary. We revised the manuscript accordingly (Page 10, Section 3.2.1).

---

### Author Rebuttal · Authors · 2026-01-23

**Rebuttal:**

We appreciate the reviewers for their constructive and thorough evaluations of our work. We are encouraged by the overall positive assessment, particularly the recognition of the novelty and strong motivation of the proposed Group-Aware Contrastive Learning (GACL) framework, the innovative use of explicit retinal morphometric features, and the impactful use of large language models to synthesize standardized clinical narratives from structured tabular data.

In response to the reviewer’s feedback, we have updated the main result tables (Tables 2, 3, and 4) to reflect improved performance gains achieved through refined hyperparameter settings and inclusion of all available eye images; these updated results remain consistent with and further strengthen the conclusions of the original manuscript. Additionally, based on the reviewers’ helpful suggestions, we incorporated additional baseline comparisons, conducted further ablation studies, and evaluated alternative method choice in GACL (replacing OR with AND) beyond our original design.

Below, we address the concerns raised by each reviewer individually and outline the improvements made in the revised manuscript. All changes made in the manuscript are highlighted in blue text.

**Supporting Material:**

/attachment/aefbe30162b5a2ddb73dec3fcd9d1f954c687d51.pdf

---

### Comment · Area_Chair_txmD · 2026-01-27
**Engage in Discussion and update score**

Dear Reviewers,

please have a thorough look at the responses by the authors. Please acknowledge the responses and engage in discussion if anything remains unclear. Currently the paper is not a clear accept. Please update your final rating by clicking “Edit” → “Official Review” and providing the Final Rating by February 1st 2026 (23:59 AoE).

Best
AC

---

### Meta-Review · Area_Chair_txmD · 2026-02-03

**Recommendation:** Accept (Poster)
**Confidence:** 4

**Metareview:**

All reviewers recommend acceptance of the paper, although two of them with "Weak Accepts". Due to the methodological innovation, the paper should be accepted.

---

### Decision · Program_Chairs · 2026-02-13

Accept (Poster)